# PG-VLM: A Multi-Stage Panoptic-Graph Architecture for Detailed Visual-Linguistic Grounding in Urban Scenes

## Abstract

Describing complex urban scenes with coherent paragraphs that are both semantically rich and spatially grounded is a key challenge for vision–language research. We present PG-VLM, a modular framework that (i) builds a Hierarchical Panoptic Scene Graph (HPSG) from panoptic segmentation, (ii) distills the graph into semantic triplets using a local instruction model, and (iii) generates narratives with a structured-to-text T5 generator. We assess text quality with standard captioning metrics and grounding with a new Narrative Relevance Detection Score (NRDS) that ties detection correctness to textual mention quality. On Cityscapes, PG-VLM surpasses recent vision-language baselines (BLIP-2, LLaVA-1.5 7B, SpatialVLM) across all metrics: CIDEr 135.0 (vs. 88.0/104.5/118.2), SPICE 28.8 (vs. 19.5/21.2/23.6), and BERTScore-F1 92.5 (vs. 88.0/89.0/90.1). Hallucination is reduced, with CHAIR-s 7.2 and CHAIR-i 9.5 (vs. 16.8/20.5 for BLIP-2, 13.0/16.2 for LLaVA-1.5, 11.4/14.8 for SpatialVLM). PG-VLM achieves substantially higher grounding via NRDS 0.76 compared to BLIP-2 at 0.52. A zero-shot check on BDD100K (50 images) indicates cross-dataset generalization (CIDEr 108.4, SPICE 24.1, NRDS-ZS 0.68), maintaining margins over all baselines. These results show that enforcing a symbolic bottleneck (HPSG to triplets) before generation improves both descriptive quality and faithfulness, offering a reproducible and extensible route to interpretable visual–language grounding in urban scenes.

## 1 Introduction

Generating paragraph-level descriptions for complex scenes remains challenging. Short captions miss object relations, spatial layout, and context typical of urban images (Ye et al., 2025; Zhang et al., 2024b; Zheng et al., 2025). Modern vision–language models (VLMs) such as BLIP-2, LLaVA, and Flamingo improve general multimodal reasoning (Zhang et al., 2024a; Yin et al., 2023) yet often produce generic, object-centric sentences with weak spatial fidelity (Singh et al., 2024; Qi et al., 2025). Even spatially aware systems including SpatialVLM (Chen et al., 2024) and ARPGrounding (Zeng et al., 2024b) struggle with occlusion, adjacency, and containment in dense layouts (Chan et al., 2023; Peng et al., 2024; Wu et al., 2024; Zhao et al., 2024a). Hallucinations remain a recurrent failure mode across model sizes and training regimes (Li et al., 2023b; Hao et al., 2025; Chang et al., 2024; Leng et al., 2024; Zhao et al., 2024b; Wang et al., 2024). Benchmarks focused on paragraph quality show that coherence and grounded relations are still limited despite large-scale pretraining (Ye et al., 2025; Zheng et al., 2025; Sarto et al., 2025; Liao et al., 2025; Li et al., 2024b; Bieri et al., 2025; Sima et al., 2024).

We present **PG-VLM**, which inserts an explicit symbolic bottleneck between vision and language. From a single urban image, Mask2Former produces panoptic outputs (Cheng et al., 2022) that we lift into a *Hierarchical Panoptic Scene Graph* (HPSG) with explicit spatial and hierarchical edges (Zhou et al., 2024; Miyanishi et al., 2023). A local instruction model converts the HPSG into canonical triplets, and a T5-based decoder maps the triplets to a multi-sentence paragraph. This design targets three pain points: spatial grounding, hallucination, and narrative flow.

On Cityscapes (Cordts et al., 2016), PG-VLM improves all automatic metrics over strong baselines (BLIP-2, LLaVA-1.5, SpatialVLM) and achieves a substantial gain on the *Narrative Relevance*

*Detection Score* (NRDS), a metric that scores whether narratively important detections are realized in text. A zero-shot check on BDD100K (Yu et al., 2020) indicates robustness to distribution shift. PG-VLM advances structured visual–linguistic grounding by describing not only what is present but also how entities relate within the scene.

## 1.1 MOTIVATION AND RESEARCH GAP

Urban driving scenes are visually dense: multiple agents share the road, static infrastructure shapes the layout, and small details such as traffic lights or signs change the interpretation of the scene. Existing vision–language models (VLMs) can recognize many object categories, but their paragraph descriptions still suffer from three recurring issues:

1. **Weak spatial grounding.** Relative positions ("in front of", "to the left of", "on the sidewalk") are often incorrect or omitted, especially in crowded layouts.

2. **Hallucinated content.** Models describe objects or events that are not supported by the image, for example an extra pedestrian or a misplaced vehicle.

3. **Limited interpretability.** End-to-end models map pixels directly to text, making it hard to inspect which visual evidence supports each sentence or to debug errors.

Prior work on detailed captioning and spatial VLMs has partially addressed these issues, for example by injecting bounding boxes into the language model or by conditioning on scene graphs. However, most methods either (i) rely on instance-only graphs without panoptic coverage of stuff regions, (ii) treat the graph as an auxiliary input rather than a true bottleneck, or (iii) evaluate mainly with surface-level metrics such as BLEU or CIDEr that do not directly reflect spatial correctness.

PG-VLM targets this gap with a multi-stage pipeline that *enforces* a symbolic bottleneck between perception and generation. A panoptic backbone builds a hierarchical panoptic scene graph (HPSG) with explicit spatial relations, a local instruction model converts the HPSG into a compact set of semantic triplets, and a T5-based decoder generates a paragraph from these triplets. This design makes it possible to (i) explicitly control which entities and relations enter the narrative, (ii) trace each sentence back to graph elements, and (iii) define an instance-level grounding metric (NRDS) that measures how well narratively important detections are realized in text.

## 2 LITERATURE REVIEW

**VLMs and paragraph generation.** Foundational VLMs couple visual encoders with language decoders and show strong zero- and few-shot performance across captioning, VQA, and retrieval (Li et al., 2023a; Alayrac et al., 2022; Koh et al., 2023; Liu et al., 2023a). They typically underperform on multi-sentence description, producing single sentences without narrative coherence or structured grounding (Ye et al., 2025; Singh et al., 2024). Long-form prompting and alignment help (Zhu et al., 2023; Yang et al., 2023) yet spatial reasoning in crowded scenes remains weak (Zhang et al., 2024a; Yin et al., 2023; Wu et al., 2024). Instruction-tuned and spatially aware models, including SpatialVLM (Chen et al., 2024), LLaVA (Liu et al., 2023c), and InstructBLIP (Dai et al., 2023), improve grounding but still miss many relations in dense layouts.

**Panoptic segmentation and scene graphs.** Transformer-based panoptic models provide strong dense predictions for both stuff and thing categories (Cheng et al., 2022). These outputs motivate panoptic scene graphs that integrate region and instance nodes with relations (Zeng et al., 2024b; Chan et al., 2023; Kim et al., 2024; Nguyen et al., 2024), sometimes aided by LLMs for relation induction (Hayder & He, 2024) or hierarchical refinement (Lee et al., 2024). We adopt an HPSG to encode spatial layout, attributes, and part–whole structure for downstream text generation (Zhou et al., 2024).

**Structured generation.** Data-to-text methods use intermediate structure such as triples or tables to improve factuality and fluency (Alaçam et al., 2024; Li et al., 2024c;b; Xue et al., 2025). T5-style encoder–decoder transformers are effective for structured-to-text mapping (Liu et al., 2023b). PG-VLM follows this recipe: HPSG to triplets to paragraph, with lightweight planning tags to improve discourse flow.

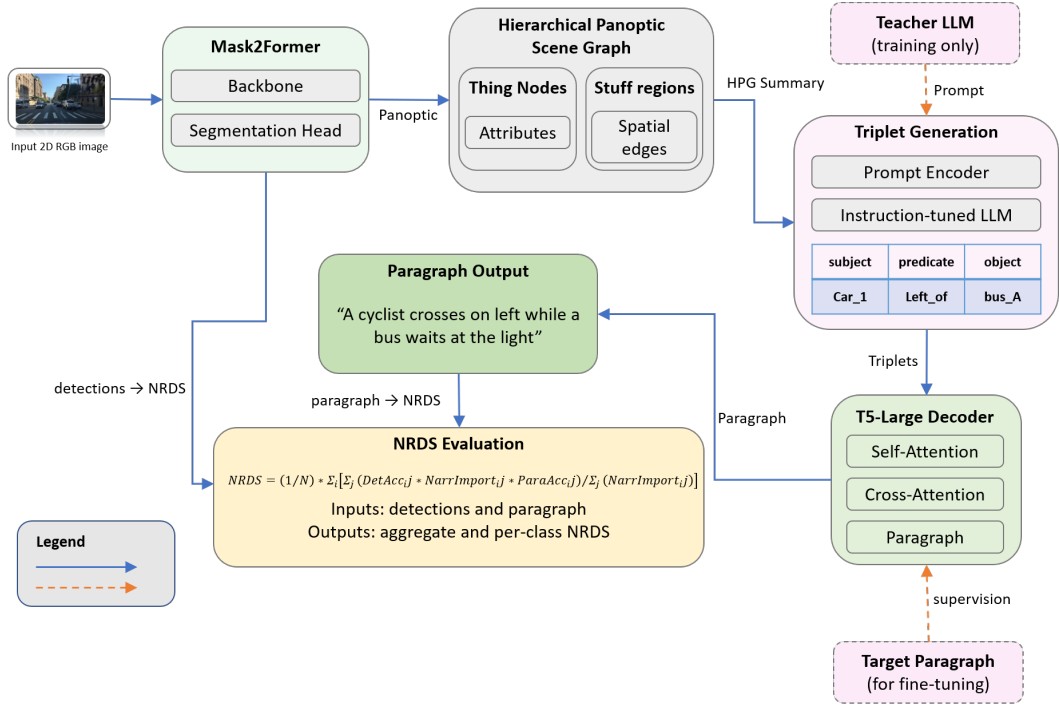

Figure 1: End-to-end PG-VLM pipeline. Mask2Former produces panoptic predictions and multi-scale features. We build an HPSG with thing and stuff nodes and explicit spatial edges, serialize salient nodes and relations, and prompt a local instruction model to produce semantic triplets. A T5-based decoder generates a paragraph from the triplets. NRDS evaluates alignment between detected entities and their narrative realization.

**Teacher supervision.** Instruction-tuned LLMs provide scalable pseudo-labels when human text is scarce (Chang et al., 2024; Yu et al., 2024; Yanuka et al., 2025; Ma et al., 2024). We use a local instruction model to synthesize paragraph targets from triplets for T5 fine-tuning, aligned with current multimodal instruction practices (Liu et al., 2023c; Dai et al., 2023; Cordts et al., 2016).

**Spatial reasoning and evaluation.** VLMs frequently mis-handle relative position, occlusion, and containment (Zheng et al., 2025; Peng et al., 2024; Zhao et al., 2024a); geometric priors or prompt design improve some cases (Dorkenwald et al., 2024; Zeng et al., 2024a; González-Chávez et al., 2023). Standard text metrics emphasize lexical overlap and can miss grounding errors (Dorkenwald et al., 2024; Sharma, 2021). Hallucination measures such as CHAIR capture unsupported mentions (Li et al., 2023b;c). We introduce NRDS to reward faithful realization of narratively important detections, complementing surface-level metrics.

## 3 METHODOLOGY: THE PG-VLM FRAMEWORK

PG-VLM converts a single RGB image into a coherent paragraph through three stages: (i) Hierarchical Panoptic Scene Graph (HPSG) construction, (ii) semantic triplet extraction with a local instruction model, and (iii) paragraph decoding with a sequence-to-sequence generator. A brief review of panoptic segmentation, scene graphs, and encoder–decoder transformers is given in Appendix A.1; here we focus on how these components are composed into our end-to-end framework. An overview appears in Fig. 1; a zoomed view of the HPSG appears in Fig. 2.

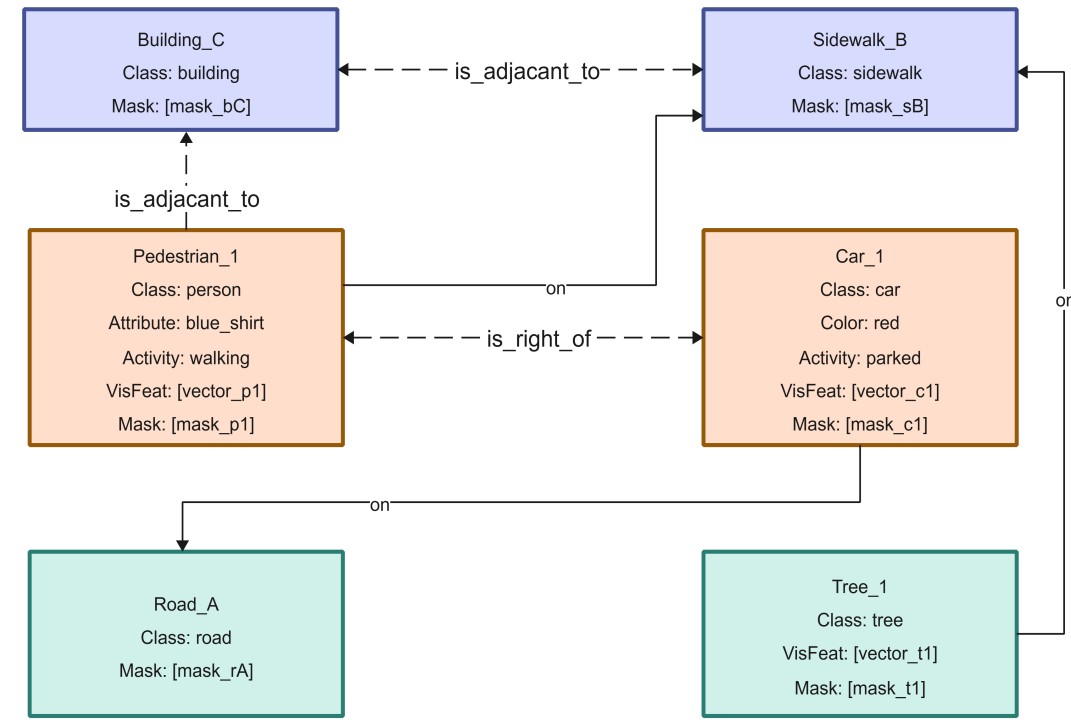

Figure 2: Zoomed view of the HPSG. Nodes store masks, boxes, features, and attributes. Edges encode spatial and hierarchical relations that drive triplet extraction and narrative planning.

## 3.1 HIERARCHICAL PANOPTIC SCENE GRAPH CONSTRUCTION

**Panoptic backbone.** Given a Cityscapes image $I$ (Cordts et al., 2016), we apply Mask2Former with a Swin-L backbone (Cheng et al., 2022). The model outputs segmentation masks $\{M_k\}_{k=1}^{K}$, boxes $\{B_k\}$, class labels $\{c_k\}$, confidence scores $\{s_k\}$, and pyramid features $\{\Phi_\ell\}$.

**Nodes.** We create nodes for all *thing* instances and *stuff* regions. For node $n$ we store: class $c(n)$, mask $M(n)$, box $B(n)$, a pooled feature

$$f(n) = \mathrm{RoIAlign}(\{\Phi_\ell\}, B(n)),$$

and optional unary attributes $\mathcal{A}(n)$ (for example color or state) derived from rule-based probes applied to $f(n)$.

**Edges.** We compute pairwise spatial relations using mask and box geometry. The predicate set is

$$\mathcal{R} = \{\texttt{on, inside, adjacent\_to, near,}^{\texttt{left\_of, right\_of, in\_front\_of, behind,}}_{\texttt{overlaps, occludes, part\_of, contacts}}\}$$

Each relation $r \in \mathcal{R}$ is scored by a differentiable geometric functional $\gamma_r(n_i, n_j)$; we keep the top-$k$ outgoing edges per node and predicate, and suppress symmetric duplicates. The resulting HPSG $G = (N, E)$ preserves layout while remaining compact for downstream text generation. A conceptual view is shown in Fig. 2.

## 3.2 SEMANTIC TRIPLET GENERATION VIA A LOCAL INSTRUCTION MODEL

**Serialization.** We linearize $G$ into a compact summary that lists salient nodes with aliases, attributes, and high-confidence edges. The summary groups facts by layout elements (road, sidewalk, building)

and by agents (car, bus, person, rider), and orders agent entries from left to right using instance centroids.

**Instruction model.** A locally hosted instruction-tuned LLM (see Section 4.5) receives the summary and emits a set of canonical *semantic triplets* $\mathcal{T} = \{(s, p, o)\}$ where $p \in \mathcal{R}$ and $s, o$ are node aliases. We constrain the predicate vocabulary to $\mathcal{R}$ and provide few-shot exemplars for format stability, following instruction-tuning practices used in LLaVA and InstructBLIP (Liu et al., 2023c; Dai et al., 2023).

**Filtering and canonicalization.** Naively forming all subject–predicate–object combinations from $G$ leads to a large and noisy set of triplets, many of which correspond to redundant or visually marginal relations. We therefore apply a two-stage filter. First, we discard triplets whose predicate is outside $\mathcal{R}$, whose arguments are not present in $G$, or whose edge score falls below a predicate-specific threshold. Symmetric predicates (e.g., `left_of`/`right_of`) are canonicalized and, when we obtain contradictory pairs, we keep the higher-confidence item.

Second, we rank the remaining triplets by a salience score that combines (i) the geometric confidence of the underlying edge, (ii) a predicate prior that favours relations such as `on`, `inside` and `in_front_of`, and (iii) the degree centrality of the incident nodes in the HPSG. We keep the top $K = 40$ triplets per image. This compact, high-confidence set $\mathcal{T}$ is then linearized into $\text{Lin}(\mathcal{T})$, which forms the input to the decoder.

### 3.3 STRUCTURED PARAGRAPH GENERATION WITH T5-LARGE

**Model and input format.** We use T5-Large (Liu et al., 2023b). The input is the ordered list of filtered triplets interleaved with lightweight planning tags:

$$\underbrace{[\text{LAYOUT}]\ (\text{road}, \texttt{adjacent\_to}, \text{sidewalk}) \cdots}_{\text{layout block}}\ \ \underbrace{[\text{AGENTS}]\ (\text{car\_1}, \texttt{on}, \text{road}) \cdots}_{\text{agent block}}$$

followed by compact unary attribute hints (for example `car_1[color=blue]`). The vocabulary includes class aliases and predicate tokens, which encourages faithful lexicalization.

**Teacher-generated targets.** For each training image, the same local instruction model is prompted with $\text{Lin}(\mathcal{T})$ to produce a target paragraph that is concise and spatially grounded. A validator removes sentences mentioning entities absent from $\mathcal{T}$, which yields consistent $(\text{Lin}(\mathcal{T}) \to P)$ pairs for supervision. This follows data-to-text planning and entity modeling best practices (Puduppully et al., 2022; 2019).

**Learning and decoding.** We minimize token-level negative log-likelihood with label smoothing 0.1. At inference we use beam search ($B = 4$) with length penalty 0.8. A constrained post-checker downranks beams that introduce unsupported entities relative to $\mathcal{T}$ while preserving fluency.

**Hyperparameters.** Inputs and targets are limited to 512 tokens. Training uses AdamW with learning rate $3 \times 10^{-5}$, weight decay 0.01, and an effective batch size of 32 via gradient accumulation, for 20 epochs. HPSG thresholds and the predicate set $\mathcal{R}$ are fixed before decoder training.

### 3.4 QUALITATIVE ILLUSTRATION

Fig. 3 presents an example image with detected instances, filtered triplets, and the final paragraph. The narrative reflects key entities and relations from the HPSG and avoids unsupported mentions.

The T5-Large decoder is trained with a standard token-level cross-entropy loss with label smoothing; full optimization and loss details are provided in Appendix A.2.

## 4 EXPERIMENTAL SETUP

We evaluate PG-VLM on Cityscapes with a zero-shot transfer check on BDD100K. This section details datasets, baselines, training configuration, and metrics.

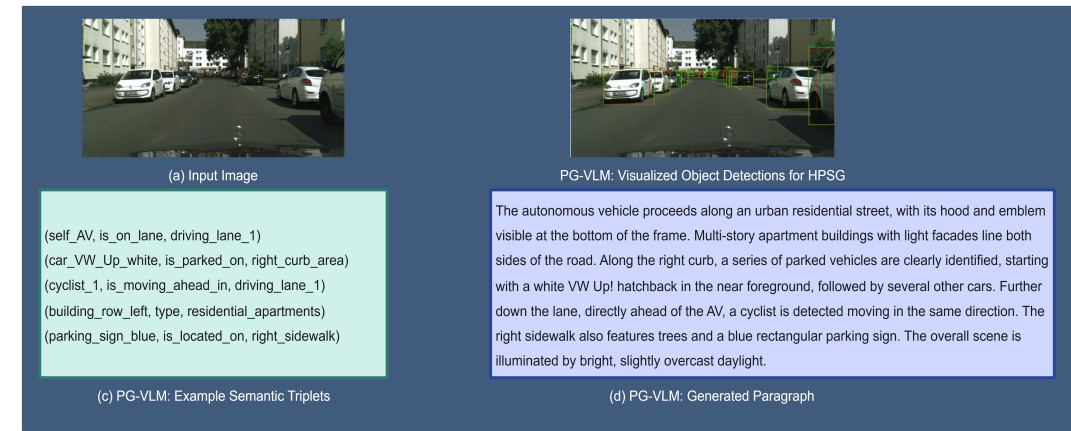

Figure 3: Qualitative example. (a) Input image, (b) detected instances and stuff regions, (c) filtered triplets, (d) generated paragraph.

## 4.1 DATASETS

**Cityscapes.** 5,000 finely annotated street-view images with 30 semantic classes (Cordts et al., 2016). We use the standard split: 2,975 train, 500 val, 1,525 test. Panoptic labels directly support HPSG construction.

**BDD100K.** A diverse driving dataset; we use a 50-image daytime subset for cross-dataset evaluation without tuning (Yu et al., 2020).

## 4.2 BASELINES

- **BLIP-2** (Li et al., 2023a): vision–language model with a Q-Former bridge.
- **LLaVA-1.5 (7B)** (Liu et al., 2023c): instruction-tuned multimodal model.
- **SpatialVLM** (Chen et al., 2024): VLM with explicit spatial grounding.

All baselines share the same preprocessing, decoding, and metric scripts.

## 4.3 TRAINING CONFIGURATION

The triplet-to-paragraph decoder is T5-Large (770M). We train with AdamW (lr $3\times10^{-5}$, weight decay $0.01$), effective batch size 32 via gradient accumulation, sequence length $512$ (input/output), for 20 epochs with early stopping on validation CIDEr. The predicate set and HPSG thresholds are fixed before training. Experiments run on a single NVIDIA RTX 4090 (24 GB).

## 4.4 EVALUATION METRICS

**Text-based metrics.** We first report standard captioning metrics: BLEU-1–4, ROUGE-L, METEOR, CIDEr, and SPICE, together with BERTScore-F1. These scores capture lexical overlap and semantic similarity between generated paragraphs and the reference texts.

**Hallucination metrics.** To assess hallucination, we use CHAIR-s/i and Entity-Precision. CHAIR-s measures the fraction of sentences that mention at least one unsupported object, while CHAIR-i measures the fraction of object mentions that are not grounded in the image. Entity-Precision counts the proportion of mentioned entities that correspond to ground-truth instances.

**Narrative Relevance Detection Score (NRDS).** The above metrics are largely text-based and do not directly test whether visually important instances are described correctly. To address this, we

use the *Narrative Relevance Detection Score* (NRDS), an instance-level grounding metric that links panoptic detections to noun phrases in the paragraph.

For a validation set of $N$ images with detection sets $\{D_i\}_{i=1}^{N}$, the overall NRDS is

$$\text{NRDS} \;=\; \frac{1}{N} \sum_{i=1}^{N} \frac{\sum_{j \in D_i} \left( \text{DetAcc}_j \cdot \text{NarrImport}_j \cdot \text{ParaAcc}_j \right)}{\text{TotalNarrImport}_i}. \tag{1}$$

Here, $\text{DetAcc}_j \in \{0,1\}$ indicates whether detection $j$ matches a ground-truth instance of the same class with IoU $> 0.5$. $\text{NarrImport}_j$ weights the narrative importance of the class of $j$; less frequent classes receive higher weight, so rare but important entities (e.g., buses, riders) contribute more. $\text{ParaAcc}_j \in [0,1]$ measures how well $j$ is realized in the paragraph: we build a short reference phrase from the HPSG attributes of $j$, find matching noun phrases in the generated text using class-specific aliases, and take the maximum CLIP similarity between the image crop of $j$ and those spans. The denominator $\text{TotalNarrImport}_i$ is the sum of narrative weights over all narratively relevant ground-truth instances in image $i$.

NRDS is related to CLIPScore but differs in two ways. CLIPScore computes a single image–paragraph similarity, whereas NRDS operates at the level of individual instances and aggregates their contributions. In addition, NRDS explicitly factors in detection correctness and class-dependent importance, so a model cannot obtain a high NRDS by describing irrelevant or incorrectly detected content. Unlike CIDEr and other text-only metrics, NRDS couples visual detection quality with the narrative and directly reflects whether important objects in the scene are mentioned accurately.

### 4.5 Training Data, Pseudo-Labels, and Evaluation Protocol

Cityscapes does not provide paragraph-level descriptions, so we adopt a teacher–student scheme for structured-to-text training. For each training image, we first build an HPSG and extract semantic triplets as described in Section 3. Unless otherwise stated, the local instruction model (the *teacher*) is `LLaMA-2-7B-Chat` (**?**), an instruction-tuned 7B LLaMA-family model run locally with 16-bit precision. We use the public checkpoint without any additional fine-tuning on Cityscapes or other driving datasets. The teacher receives the triplets and produces a concise, spatially grounded paragraph; an automatic validator removes sentences that mention entities not present in the input triplets. The resulting paragraphs serve as *pseudo-labels* for the T5 decoder and as reference texts for all models (PG-VLM and baselines) when computing automatic captioning metrics (CIDEr, SPICE, BERTScore, BLEU, ROUGE-L, METEOR). Baselines (BLIP-2, LLaVA-1.5, SpatialVLM) are evaluated in their publicly released form without any fine-tuning on these references.

This protocol can introduce a bias in favour of PG-VLM, since the pseudo-labels reflect the teacher's style and the structure of the triplets. To partially control for this, we (i) report hallucination metrics (CHAIR-s/i, Entity-Precision) and NRDS, which depend on image–text alignment rather than lexical similarity to the teacher, and (ii) conduct a blind human evaluation in which annotators compare model outputs given the *image* rather than the teacher text; results are reported in Appendix A.8.

## 5 Results

We report automatic metrics on Cityscapes and a cross-dataset check on BDD100K. PG-VLM is compared against SpatialVLM, LLaVA-1.5, and BLIP-2 under identical evaluation settings.

### 5.1 Automatic Evaluation on Cityscapes

PG-VLM yields consistent gains across n-gram overlap (BLEU), long-form coverage (ROUGE-L), semantic adequacy (METEOR, SPICE, BERTScore), and consensus-based relevance (CIDEr). Improvements are most pronounced in CIDEr and SPICE, which are sensitive to content selection and relational details. The pattern suggests that the HPSG→triplet bottleneck improves both *what* is said (entity/attribute coverage) and *how* it is organized (coherent, multi-sentence narratives). Qualitatively, we observe better grounding of spatial predicates and fewer missing scene elements, which matches the larger margins on SPICE and BERTScore.

Table 1: Automatic evaluation of paragraph generation on Cityscapes. Higher is better.

| Metric | PG-VLM | SpatialVLM | LLaVA-1.5 | BLIP-2 |
|--------|--------|------------|-----------|--------|
| BLEU-1 | 78.2 | 72.0 | 69.5 | 65.1 |
| BLEU-2 | 63.5 | 57.8 | 54.2 | 48.8 |
| BLEU-3 | 51.0 | 45.6 | 42.1 | 36.0 |
| BLEU-4 | 40.8 | 36.0 | 32.8 | 27.3 |
| ROUGE-L | 60.5 | 55.2 | 52.4 | 49.2 |
| METEOR | 31.5 | 28.1 | 26.3 | 24.0 |
| CIDEr | 135.0 | 118.2 | 104.5 | 88.0 |
| SPICE | 28.8 | 23.6 | 21.2 | 19.5 |
| BERTScore (F1) | 92.5 | 90.1 | 89.0 | 88.0 |

Table 2: Hallucination and entity grounding on Cityscapes. Lower is better for CHAIR; higher is better for Entity-Precision.

| Metric | PG-VLM | SpatialVLM | LLaVA-1.5 | BLIP-2 |
|--------|--------|------------|-----------|--------|
| CHAIR-s $\downarrow$ | 7.2 | 11.4 | 13.0 | 16.8 |
| CHAIR-i $\downarrow$ | 9.5 | 14.8 | 16.2 | 20.5 |
| Entity-Precision $\uparrow$ | 89.6 | 84.2 | 81.5 | 77.0 |

## 5.2 HALLUCINATION AND ENTITY GROUNDING

PG-VLM reduces unsupported entity mentions (CHAIR-s/i) while improving correct mentions (Entity-Precision). This trend aligns with the model's explicit symbolic bottleneck: triplets constrain lexical realization to entities and relations present in the graph, curbing free-form drift and yielding stronger image–text fidelity.

## 5.3 NARRATIVE RELEVANCE DETECTION SCORE

NRDS measures whether narratively important visual entities are realized faithfully in text (Section 4.4). On Cityscapes validation, PG-VLM achieves **0.76** versus **0.52** for BLIP-2, indicating better alignment between detection, narrative salience, and paragraph realization. We observe a strong correlation between NRDS and CIDEr across samples, while cases with high CIDEr but low NRDS typically omit salient instances or introduce subtle hallucinations—scenarios where PG-VLM shows fewer failures.

## 5.4 CROSS-DATASET GENERALIZATION

PG-VLM maintains an advantage without dataset-specific tuning. The HPSG→triplet abstraction helps preserve core semantics and spatial relations under shifts in camera geometry and urban context, supporting generalization beyond the training distribution.

## 6 ABLATIONS AND ANALYSIS

We conduct controlled experiments on Cityscapes validation to isolate contributions of each PG-VLM component, keeping training settings constant with the main model.

## 6.1 EFFECT OF THE HPSG BOTTLENECK

Removing the graph stage and feeding visual features directly to the decoder reduces content selection quality (CIDEr), relational coverage (SPICE), and faithfulness (NRDS), while doubling unsupported mentions (CHAIR). The symbolic bottleneck forces early disambiguation of entities and relations, which translates into more reliable downstream text.

Table 3: Zero-shot transfer to BDD100K daytime subset (50 images).

| Model | CIDEr ↑ | SPICE ↑ | NRDS-ZS ↑ |
|---|---|---|---|
| PG-VLM | 108.4 | 24.1 | 0.68 |
| SpatialVLM | 95.7 | 20.3 | 0.57 |
| LLaVA-1.5 (7B) | 88.9 | 18.4 | 0.51 |
| BLIP-2 | 73.2 | 16.2 | 0.42 |

Table 4: Impact of the structured HPSG bottleneck. Removing HPSG weakens spatial grounding and increases hallucination.

| Model Variant | CIDEr ↑ | SPICE ↑ | CHAIR-s ↓ | NRDS ↑ |
|---|---|---|---|---|
| PG-VLM (full) | 135.0 | 28.8 | 7.2 | 0.76 |
| Direct ViT $\rightarrow$ T5 | 112.3 | 22.5 | 13.4 | 0.59 |

## 6.2 TRIPLET FILTERING AND SALIENCE

Restricting the input to high-confidence, non-redundant triplets improves precision and NRDS. The decoder benefits from a concise fact plan rather than an over-complete set that can encourage repetition or off-target elaboration.

## 6.3 DECODER VARIANTS AND PLANNING SIGNALS

We compare decoder scales and planning tags in Appendix A.5, showing that lightweight layout-first, left-to-right agent ordering improves ROUGE-L, SPICE, and NRDS without adding trainable planners.

## 6.4 SENSITIVITY TO PREDICATE INVENTORY AND TRIPLET BUDGET

We vary the predicate inventory and the triplet budget $K$. Extremely small inventories collapse distinct spatial cues (e.g., merging `in_front_of` with `left_of`), reducing SPICE and NRDS. Increasing $K$ beyond the chosen setting offers little benefit and can slightly degrade fluency due to verbosity. The default inventory and $K$ strike a practical balance between coverage and concision.

## 6.5 QUALITATIVE ANALYSIS

Qualitative inspection (see Figure 3) shows that PG-VLM consistently realizes spatial relations (e.g., *car beside bus*, *pedestrian on sidewalk*) and avoids unsupported mentions that appear in end-to-end baselines. We also observe better narrative cohesion across sentences, aided by triplet ordering and grouping.

## 7 CONCLUSION

PG-VLM introduces a structured pipeline for paragraph-level scene understanding that combines panoptic segmentation, a hierarchical panoptic scene graph, semantic triplets, and a sequence-to-sequence decoder. By grounding generation in explicit symbolic facts, PG-VLM improves spatial fidelity and reduces hallucination compared with recent vision–language models. On Cityscapes, it delivers consistent gains across captioning and semantic metrics and lowers hallucination rates, while maintaining an advantage under zero-shot transfer to BDD100K. The Narrative Relevance Detection Score complements traditional metrics by emphasizing alignment between detected entities and their textual realization. Altogether, the results support the value of an HPSG→triplet bottleneck for reliable, coherent narratives of complex urban scenes.

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

## A  APPENDIX

### A.1  PRELIMINARIES

This section summarizes the standard building blocks used by PG-VLM: panoptic segmentation for dense visual recognition, scene graphs for structured reasoning, and encoder–decoder transformers for text generation. The goal is to fix notation and clarify how these components interface in our pipeline.

**Panoptic segmentation.** Panoptic segmentation assigns a semantic label to every pixel while distinguishing instance identities for countable categories. Let $I \in \mathbb{R}^{H \times W \times 3}$ be an input image. A panoptic model produces $(\boldsymbol{M}, \boldsymbol{Y})$, where $\boldsymbol{M} = \{M_k\}_{k=1}^{K}$ are binary masks and $\boldsymbol{Y} = \{(c_k, s_k)\}_{k=1}^{K}$ are per-instance class labels $c_k$ and scores $s_k$. Stuff regions such as road or sidewalk are represented as single masks without an instance index. We use Mask2Former with a Swin backbone (Cheng et al., 2022) to obtain masks, boxes, and multi-scale features that seed graph construction. In PG-VLM, these panoptic outputs are the only direct interface between raw pixels and the symbolic scene graph.

**Scene graph representations.** A scene graph is a directed multigraph $G = (N, E)$ with node set $N$ and labeled edges $E \subseteq N \times \mathcal{R} \times N$, where $\mathcal{R}$ is a predicate set. In a panoptic scene graph (Li et al., 2024a), nodes include both *thing* instances (cars, pedestrians, riders) and *stuff* regions (road, sidewalk, building). For a node $n \in N$, we store $(c(n), M(n), B(n), f(n))$: semantic class, mask, bounding box, and a pooled visual feature. PG-VLM constructs a *Hierarchical Panoptic Scene Graph*

(HPSG) in which edges cover spatial relations (e.g., left_of, on, in_front_of), containment and part–whole links, and selected unary attributes as key–value tags. The HPSG serves as the sole structured input to the language stack and provides explicit hooks for reasoning about layout and interactions.

**Sequence-to-sequence models.** Given a linearized structured input $X_{\text{struct}}$ and a target paragraph $P = (s_1, \ldots, s_T)$, an encoder–decoder transformer models

$$p(P \mid X_{\text{struct}}) = \prod_{t=1}^{T} p(s_t \mid s_{<t}, X_{\text{struct}}).$$

We instantiate the decoder with T5-Large (Liu et al., 2023b) and format $X_{\text{struct}}$ as a sequence of canonical triplets plus compact attribute hints. This separation of perception (panoptic segmentation) from generation (seq2seq) allows the model to focus on content selection and discourse structure while preserving explicit grounding hooks through the triplets. In particular, entities and relations that are not present in the HPSG cannot enter the input sequence, which constrains the decoder and reduces free-form hallucination.

A.2 Reproducibility and Implementation Details

**Environment.** PyTorch 2.2, CUDA 12.x, Transformers 4.41, Accelerate 0.30, SentencePiece 0.1.99. Mixed precision (bfloat16) for all model stages. Fixed global seed 1234 for data shuffling, sampler, and decoder initialization.

**Preprocessing.** Cityscapes frames are resized to $1024 \times 512$ (longer side 1024), normalized to backbone statistics, and batched by image area. Panoptic maps use the official evaluation format; category aliases follow the dataset label policy.

**HPSG construction.** Mask2Former (Swin-L) outputs are filtered at confidence $\geq 0.5$ for *thing* instances. Edges are created from masks and boxes: on, inside, left_of, right_of, in_front_of, behind, adjacent_to, near, overlaps, occludes, part_of, contacts. We apply per-predicate non-maximum suppression using IoU on relation supports (union of incident masks) and retain top-$k = 6$ outgoing edges per node per predicate.

**Triplet serializer and filter.** Nodes are serialized with {id, class, mask-derived attributes (e.g., color hue bin), size bin, centroid}. We keep up to $K = 40$ triplets ranked by (predicate prior, edge confidence, degree centrality). Symmetric duplicates (left_of/right_of) are canonicalized; cycles on on/inside are pruned.

**Decoder training.** T5-Large; AdamW ($3 \times 10^{-5}$, weight decay 0.01), label smoothing 0.1, batch size 32 (via gradient accumulation), 512 tokens input/output, 20 epochs with early stopping on validation CIDEr. Beam search size 4, length penalty 0.8. A constrained post-checker downranks beams with entities not present in the input triplets.

**Training objective.** The teacher LLM is kept fixed; we do not fine-tune it on Cityscapes. Its role is to (i) generate semantic triplets from HPSG summaries during training-time data preparation and (ii) synthesize target paragraphs from those triplets. The learnable component in PG-VLM is the T5-Large decoder. Given an input sequence $\text{Lin}(\mathcal{T})$ of filtered triplets and a target paragraph $P$, the model is trained with a standard token-level cross-entropy loss

$$\mathcal{L} = -\sum_{t=1}^{T} \log p_\theta(s_t \mid s_{<t}, \text{Lin}(\mathcal{T})),$$

with label smoothing 0.1. We use AdamW with learning rate $3 \times 10^{-5}$, weight decay 0.01, and an effective batch size of 32. NRDS is used only for evaluation and model selection; it does not appear in the loss.

### A.3 PROMPT TEMPLATES

**Triplet extraction (HPSG→Triplets).**

> **System:** You receive a scene summary with nodes and relations. Return a list of triplets in the canonical form (subject, predicate, object) using this predicate set: on, inside, left_of, right_of, in_front_of, behind, adjacent_to, near, overlaps, occludes, part_of, contacts. Use node ids (e.g., car_3, road_1). Do not invent objects.
> **User:** NODES: {...} EDGES: {...}
> **Assistant:** (car_3, on, road_1); (person_5, near, crosswalk_1); ...

**Paragraph teacher (Triplets→Paragraph).**

> **System:** Write a concise paragraph (3–6 sentences) describing the scene using only the provided triplets and attributes. Preserve spatial relations and avoid objects not listed. Maintain a natural flow: layout first, then agents left to right.
> **User:** TRIPLETS: {...} ATTRS: {...}
> **Assistant:** *The road stretches ahead with buildings on both sides...*

### A.4 NARRATIVE RELEVANCE DETECTION SCORE (NRDS)

Standard captioning metrics emphasize lexical overlap and often miss whether narratively important visual entities are faithfully realized in text. NRDS quantifies visual–text alignment by combining detection correctness, class-level narrative importance, and paragraph realization.

For a validation set of $N$ images with detection sets $\{D_i\}_{i=1}^N$, the overall NRDS is

$$\text{NRDS} = \frac{1}{N}\sum_{i=1}^{N}\frac{\sum_{j\in D_i}\left(\text{DetAcc}_j \cdot \text{NarrImport}_j \cdot \text{ParaAcc}_j\right)}{\text{TotalNarrImport}_i}. \tag{2}$$

**Components.**

- $D_i$ is the set of detected instances for image $i$.
- $\text{DetAcc}_j \in \{0,1\}$ is 1 if detected instance $j$ matches a ground-truth instance of the same class with IoU $> 0.5$, else 0.
- $\text{NarrImport}_j$ weights class importance. Let $c$ be the class of instance $j$ and $F_c$ the number of class-$c$ instances in training. We set $\text{NarrImport}_j = 1/\sqrt{F_c}$ to emphasize less frequent classes without extreme weights.
- $\text{ParaAcc}_j \in [0,1]$ measures textual realization. We construct a short reference phrase from HPSG attributes/relations for $j$, locate candidate spans via class aliases and noun-phrase chunking, compute CLIP cosine similarity between the image crop of $j$ and each span, take the maximum, and normalize to $[0,1]$.
- $\text{TotalNarrImport}_i$ is the sum of $\text{NarrImport}_j$ over narratively relevant ground-truth instances in image $i$.

**Implementation details.** Detections and ground truth are matched per class with one-to-one assignment. Crops are taken from instance masks with small padding; class aliases include common synonyms for Cityscapes classes. We report aggregate NRDS and per-class NRDS.

### A.5 ADDITIONAL QUANTITATIVE RESULTS

Table 6: Per-class NRDS for key *thing* categories. Higher is better.

| Class | Car | Person | Bicycle | Bus | Truck | Rider | Motorcycle | Train |
|---|---|---|---|---|---|---|---|---|
| NRDS | 0.82 | 0.73 | 0.65 | 0.80 | 0.60 | 0.58 | 0.55 | 0.75 |

**Per-class NRDS on Cityscapes (validation).**

Table 7: Paragraph length control with beam search. Tokens approximate mean output length.

| Setting | Tokens | CIDEr ↑ | SPICE ↑ | CHAIR-s ↓ |
|---|---|---|---|---|
| Short (minlen) | 85 | 126.7 | 27.1 | 8.0 |
| Default | 120 | 135.0 | 28.8 | 7.2 |
| Long (no cap) | 165 | 133.9 | 28.6 | 7.8 |

**Length control versus quality.**

Table 8: Effect of ordering/grouping on discourse and grounding.

| Input Order | ROUGE-L ↑ | SPICE ↑ | CHAIR-s ↓ | NRDS ↑ |
|---|---|---|---|---|
| Shuffled | 58.9 | 27.4 | 8.1 | 0.72 |
| Layout→Agents (ours) | 60.5 | 28.8 | 7.2 | 0.76 |

**Triplet ordering and grouping.**

**CIDEr–NRDS correlation.** Across validation samples, Pearson $r = 0.88$, indicating strong alignment. Outliers with high CIDEr but lower NRDS typically omit a salient entity or conflate two adjacent agents, which the symbolic bottleneck mitigates.

## A.6 ROBUSTNESS ANALYSES

**Occlusion sensitivity.** We bin instances by occlusion ratio (mask overlap with other instances).

Table 9: NRDS versus occlusion level.

| Occlusion bin | $[0, 0.2)$ | $[0.2, 0.5)$ | $[0.5, 1.0]$ |
|---|---|---|---|
| NRDS | 0.79 | 0.74 | 0.66 |

**Lighting/contrast perturbation.** We apply uniform brightness/contrast jitters at inference time (no retraining).

## A.7 RUNTIME AND MEMORY

Throughput is ∼4.1 FPS in steady state with batched Mask2Former inference. VRAM peaks when panoptic features and the decoder are both resident; gradient-free inference limits memory pressure.

## A.8 HUMAN EVALUATION PROTOCOL

Three annotators evaluated 150 validation images (balanced across density). Each paragraph was scored on 1–5 Likert scales for *Fluency*, *Relevance*, and *Spatial correctness*. Items were randomized per annotator; model identities were blinded. Krippendorff's $\alpha = 0.62$ averaged across dimensions.

## A.9 ERROR ANALYSIS

- **Small-instance miss:** rare classes (e.g., *rider*, *motorcycle*) missed under heavy occlusion can lead to under-reporting. Mitigation: predicate smoothing and per-class triplet priors.
- **Attribute drift:** color/state mismatches for traffic lights at distance. Mitigation: attribute confidence gating and span-level fallback.
- **Relation ambiguity:** camera parallax confuses `in_front_of` vs. `left_of`. Mitigation: tie-break by orientation priors and vanishing-point heuristics.

Table 10: NRDS under test-time brightness/contrast jitter.

| Perturbation | None | Mild | Strong |
|---|---|---|---|
| NRDS | 0.76 | 0.74 | 0.70 |

Table 11: Per-image inference on RTX 4090 (24GB) at $1024 \times 512$.

| Stage | Latency (ms) | Peak VRAM (GB) |
|---|---|---|
| Mask2Former panoptic | 85 | 9.2 |
| HPSG build (CPU+GPU mix) | 15 | 0.4 |
| Triplet LLM (local) | 90 | 2.1 |
| T5-Large decoding (beam=4) | 55 | 3.1 |
| **Total** | **245** | **14.8** |

- **Noun-phrase alignment collisions:** aliases overlapping with other classes (e.g., *rider/person*) reduce ParaAcc. Mitigation: class-conditional lexicons.

## A.10 POST-CHECKER AND RE-RANKING DETAILS

The post-checker computes entity coverage $\mathcal{C}$ and unsupported mentions $\mathcal{U}$ per candidate beam. The final score is

$$s_{\text{final}} = s_{\text{beam}} + \lambda_{\text{cov}} \cdot \mathcal{C} - \lambda_{\text{unsup}} \cdot \mathcal{U},$$

with $\lambda_{\text{cov}} = 0.6$, $\lambda_{\text{unsup}} = 0.8$. $\mathcal{C}$ counts unique input entities realized at least once; $\mathcal{U}$ counts span-level entities not in the input triplets (exact/alias match). This re-ranking consistently reduces CHAIR while preserving fluency.

## A.11 DATASET AND LICENSING NOTES

Cityscapes (Cordts et al., 2016) and BDD100K (Yu et al., 2020) are used under their respective academic licenses. Faces and license plates in figures are blurred per dataset policy. No personal data is newly collected.

## A.12 BROADER IMPACT AND RISK CONTROLS (EXTENDED)

We include a brief operational guidance: require minimum thresholds on CHAIR and NRDS before any deployment; maintain human-in-the-loop review; log coverage and confidence per paragraph; apply privacy filters to sensitive details; continuously evaluate across diverse cities, weather, and times of day.

## A.13 ADDITIONAL QUALITATIVE EXAMPLES

We provide more end-to-end examples (image, HPSG snippet, triplets, paragraph) to illustrate typical successes and remaining challenges. See Figure 3 in the main paper and the supplemental gallery provided with the code release.

**Limitations and future work.** PG-VLM depends on the quality of the panoptic backbone and the teacher LLM: severe panoptic failures or biased teacher paragraphs can propagate to the triplets and the final narrative. Our evaluation is also centered on urban driving scenes (Cityscapes and a small BDD100K subset), so generalization to other domains remains to be tested. Finally, NRDS, while more grounded than text-only metrics, still relies on CLIP and heuristic phrase matching. Extending PG-VLM and NRDS to a broader set of datasets and detection backbones is an important direction for future work.

**Code and model release.** Upon publication we will release the full PG-VLM codebase, including scripts for HPSG construction, triplet extraction, paragraph generation, and NRDS computation. We

Table 12: Mean human ratings (1–5). Higher is better.

| Model | Fluency | Relevance | Spatial Corr. |
|---|---|---|---|
| PG-VLM | 4.3 | 4.1 | 4.0 |
| SpatialVLM | 3.9 | 3.7 | 3.6 |
| LLaVA-1.5 | 3.8 | 3.6 | 3.4 |
| BLIP-2 | 3.6 | 3.4 | 3.1 |

will also release configuration files, pretrained weights for the T5-Large decoder, and evaluation scripts used to obtain all reported numbers. Where licenses permit, we will additionally provide the panoptic backbone checkpoint and teacher- generated training paragraphs for Cityscapes to facilitate exact reproduction of our experiments.

