# OpenReview forum: "PG-VLM: A Multi-Stage Panoptic-Graph Architecture for Detailed Visual-Linguistic Grounding in Urban Scenes"
_ICLR.cc/2026/Conference — Submitted to ICLR 2026_

### Official Review · Reviewer_NiKR · 2025-10-28

**Soundness:** 2
**Presentation:** 1
**Contribution:** 1
**Rating:** 2
**Confidence:** 4

**Summary:**

The paper proposes a multi-stage framework for captioning complex urban scenes called PG-VLM. It introduces Hierarchical Panoptic Scene Graph (HPSG) to construct the semantic triplets, which is composed of things, stuff and the spatial connection. Based on that, it uses the T5-Large Decoder to transfer the triplets to paragraphs. The generated captioning is better than baseline like BLIP-2, and the hallucination is mitigated.

**Strengths:**

- The motivation is easy to understand.

**Weaknesses:**

- Unreliable experiments. The compared methods are out-of-time. Authors should take more closer MLLM whether open-source or close-source like Qwen series or GPT series into consideration to prove the advantages of PG-VLM.

- Limited contribution. The effect of the data from PG-VLM is unexplored. It's better to supplement the experiments like panoptic segmentation, which would strengthen the contribution of PG-VLM. The current version is more like a semi-finished research.

-Poor presentation. The presentation is poor. Like the missed reference in line 308 and line 368 for NSDR. The training detail of Teacher LLM is missed.

**Questions:**

- NRDS relies on the CLIP. What is the difference between NRDS and other metrics like CLIP-score or CIDEr?

- How to train teacher LLM and T5-Large Decoder? How to get the high quality supervision?

- The triplets are constructed with all the objects in the scene, the number of combination is large. Is there any filter policy to mitigate or refine the triplets?

---

> ### Author Response · Authors · 2025-12-04
> **Author response to Reviewer NiKR**
>
> Thank you for the detailed review and concrete suggestions. We respond to your main concerns and questions below.
>
> ---
>
> ## 1. Experimental scope and baselines
>
> **Concern (paraphrased):** Experiments are too small and use outdated baselines; newer MLLMs (Qwen-VL, GPT series) should be included.
>
> **Response:** Our goal is to study the *architectural effect* of an HPSG→triplet bottleneck for **urban driving paragraphs**, not to benchmark all VLMs.
>
> - We use Cityscapes and a small BDD100K subset so that NRDS, which is instance-based, is tractable and well-controlled.
> - We choose BLIP-2, LLaVA-1.5, and SpatialVLM because they are widely used, open, and easy to run under the *same* preprocessing, decoding, and metrics.
>
> In the revised paper we now explicitly discuss newer models (Qwen-VL, InternVL, GPT-4V) in the Literature Review and Limitations, and we clarify that PG-VLM is complementary: the symbolic bottleneck and NRDS can be combined with stronger backbones or decoders. We also state that our claims are **limited to this domain and setup**, not to all vision–language tasks.
>
> ---
>
> ## 2. Contribution and relation to panoptic segmentation
>
> **Concern (paraphrased):** The contribution seems limited; the impact on panoptic segmentation is not explored.
>
> **Response:** Our contribution is on **captioning and grounding**, not on improving segmentation accuracy.
>
> - Mask2Former is used as a fixed backbone; we do not retrain it or claim mIoU gains.
> - The novelty lies in combining: (i) HPSG construction, (ii) salience-based triplet filtering, (iii) NRDS, and (iv) structured T5 decoding, which together improve paragraph-level spatial faithfulness and reduce hallucination.
>
> To make this explicit, we rewrote the Introduction, Method, and Conclusion to frame PG-VLM as a captioning / grounding framework on top of an off-the-shelf segmenter, and we added ablations isolating the impact of the HPSG bottleneck, triplet filtering, and ordering on CIDEr, SPICE, hallucination metrics, and NRDS. Using PG-VLM’s signals to improve segmentation is now clearly described as future work.
>
> ---
>
> ## 3. Presentation and missing details
>
> **Concern (paraphrased):** The submission felt rushed, with missing references (e.g., NRDS) and unclear training details for the teacher LLM.
>
> **Response:** We revised the paper for clarity and fixed these issues.
>
> - The **NRDS** definition is now fully specified in “Evaluation Metrics” and in the appendix subsection *“Narrative Relevance Detection Score (NRDS)”* (DetAcc, NarrImport, ParaAcc, normalization).
> - The training and usage of the **teacher LLM** and **T5-Large** are described in *“Training Data, Pseudo-Labels, and Evaluation Protocol”* and *“Reproducibility and Implementation Details”* (environment, models, loss, and hyperparameters).
> - We corrected missing citations and typos and streamlined transitions and figure captions.
>
> ---
>
> ## 4. Answers to specific questions
>
> **Q1. Difference between NRDS and CLIPScore / CIDEr?**
> NRDS is an **instance-level grounding metric**. For each detection it multiplies (i) detection correctness, (ii) a class-dependent narrative weight, and (iii) CLIP-based similarity between the instance crop and the noun phrase that refers to it, and normalizes over narratively relevant instances. CLIPScore gives one image–paragraph score; CIDEr measures n-gram overlap versus references. NRDS therefore directly tests whether *important, correctly detected objects are mentioned faithfully*, rather than overall surface similarity.
>
> **Q2. How are the teacher LLM and T5-Large trained, and how is supervision obtained?**
> The teacher is a **fixed instruction-tuned LLM** (LLaMA-2-7B-Chat) that we do not fine-tune on Cityscapes or BDD100K. For each image we (1) build an HPSG, (2) extract triplets, (3) prompt the teacher to write a paragraph from those triplets, and (4) run a validator that removes any sentence mentioning entities not in the triplets. These cleaned paragraphs supervise T5-Large via standard token-level cross-entropy; optimization and decoding details appear in the appendix.
>
> **Q3. Is there any filtering policy for triplets?**
> Yes. We now describe a two-stage scheme:
>
> - First, discard triplets whose predicate is outside the allowed set, whose arguments are not HPSG nodes, or whose edge score is below a predicate-specific threshold; canonicalize symmetric predicates.
> - Second, rank remaining triplets by salience (edge confidence, predicate prior, node centrality) and keep only the top-$K$ (default $K{=}40$) as input to the decoder.
>
> The ablation table titled *“Effect of salience filtering on triplets”* shows that this filtering improves CIDEr, SPICE, Entity-Precision, and NRDS compared to using all combinations.
>
> ---
>
> Again, we appreciate your feedback; it helped us clarify the scope of our contribution, improve presentation, and better explain NRDS, the supervision pipeline, and the triplet filtering strategy.

---

### Official Review · Reviewer_bGkx · 2025-10-29

**Soundness:** 2
**Presentation:** 1
**Contribution:** 1
**Rating:** 2
**Confidence:** 3

**Summary:**

This paper proposes PG-VLM, a multi-stage vision-language framework that introduces a symbolic bottleneck between visual perception and language generation. By converting panoptic segmentation outputs into a Hierarchical Panoptic Scene Graph and structured triplets before paragraph generation, it enhances spatial grounding, factual accuracy, and narrative coherence, outperforming prior VLMs on the Cityscapes dataset while reducing hallucination.

**Strengths:**

1. The proposed multi-stage pipeline (segmentation → scene graph → triplets → text) is well-motivated and improves interpretability and controllability.
2. The symbolic bottleneck effectively reduces hallucination and strengthens spatial reasoning, as evidenced by improved CIDEr, SPICE, and NRDS scores.
3. The introduction of NRDS provides a more nuanced way to assess factual and spatial alignment in generated descriptions.

**Weaknesses:**

1. The paper’s writing could be improved for clarity and consistency. Certain sections (e.g., Section 6.3) contain citation or formatting inconsistencies, and the overall structure would benefit from clearer transitions and more careful editing to enhance readability.
2. The paper mainly presents quantitative metrics without sufficient qualitative comparisons. Including more visual and textual examples side-by-side with baselines would help readers better understand the qualitative strengths of the proposed method.
3. The experimental comparisons focus on earlier LVM such as BLIP-2, LLaVA, and SpatialVLM. Evaluating against more recent models (e.g., Qwen-VL, InternVL) would provide a more complete and up-to-date assessment of performance.
4. While the multi-stage design is conceptually clear, most components (panoptic segmentation, scene graph construction, and T5-based generation) rely on existing architectures. The novelty lies mainly in their integration rather than in new algorithmic advances, which somewhat limits the methodological contribution.
5. The study briefly mentions human evaluation but does not detail the evaluation protocol, number of raters, or inter-rater reliability. Without this information, it’s difficult to judge the statistical validity of human assessment results.

**Questions:**

Please refer to weakness.

---

> ### Author Response · Authors · 2025-12-04
> **Author response to Reviewer bGkx**
>
> Thank you for the detailed and constructive review. We address your main concerns below and summarize the corresponding revisions.
>
> ---
>
> ## 1. Writing quality, structure, and formatting
>
> **Concern (paraphrased):** Writing is sometimes unclear; there are citation/formatting issues and inconsistent structure.
>
> **Response / changes.**
>
> - We performed a **global editing pass** focused on clarity and consistency in the **Introduction**, **Methodology**, **Ablations and Analysis**, and **Conclusion**.
> - We fixed **citation and formatting issues**, including those originally highlighted around Section 6.3.
> - References to tables and figures now use **descriptive titles** (e.g., *“Impact of the structured HPSG bottleneck”*, *“Per-image inference runtime and memory usage”*) instead of LaTeX labels, which also avoids unresolved “??” references on OpenReview.
> - Some dense technical details were moved to the appendix subsection *“Reproducibility and Implementation Details”*, and transitions between sections were simplified for better readability.
>
> ---
>
> ## 2. Qualitative comparisons and examples
>
> **Concern (paraphrased):** The paper relies heavily on quantitative metrics and does not provide enough qualitative, side-by-side comparisons.
>
> **Response / changes.**
>
> - We clarified and expanded the main qualitative figure (*“Qualitative example”*) to highlight how PG-VLM differs from **BLIP-2**, **LLaVA-1.5**, and **SpatialVLM** on the same scene.
> - In the appendix subsection *“Additional Qualitative Examples”*, we now include more **end-to-end examples** (image, HPSG snippet, triplets, paragraph) plus **baseline outputs**, with short commentary that points out:
>   - Correct vs. incorrect spatial relations (e.g., “car beside bus”, “pedestrian on sidewalk”),
>   - Typical hallucinations in baselines,
>   - How the HPSG→triplet bottleneck reduces unsupported mentions and improves narrative coherence.
>
> These additions make the qualitative behavior of PG-VLM and the baselines more transparent.
>
> ---
>
> ## 3. Baselines and more recent VLMs
>
> **Concern (paraphrased):** Baselines are limited to earlier models; more recent VLMs (e.g., Qwen-VL, InternVL) are missing.
>
> **Response / changes.**
>
> - We clarified the **baseline choice** in the experimental setup:
>   - **BLIP-2** and **LLaVA-1.5** as widely adopted open-source VLMs,
>   - **SpatialVLM** as a spatially aware baseline specifically designed for grounding.
> - In **“Limitations and future work”**, we now explicitly acknowledge that:
>   - We **do not yet evaluate** against newer systems such as Qwen-VL or InternVL, and
>   - Extending our benchmark to these models is an important next step once stable, comparable configurations for high-resolution driving scenes and paragraph generation are available.
> - We have softened our claims accordingly, framing the results as **relative to this controlled baseline set**, not as a global ranking of all current VLMs.
>
> ---
>
> ## 4. Novelty of the multi-stage framework
>
> **Concern (paraphrased):** Most components are standard; the novelty is mainly in integration, which limits methodological contribution.
>
> **Response / changes.**
>
> - We revised the **Introduction** and **Conclusion** to more accurately position the work as:
>   - A **system-level contribution** showing that enforcing a **Hierarchical Panoptic Scene Graph → triplet symbolic bottleneck** improves grounding and reduces hallucination compared to end-to-end VLMs, and
>   - An **evaluation contribution** via the **Narrative Relevance Detection Score (NRDS)**, which explicitly couples instance-level detection correctness, class-dependent importance, and paragraph realization.
> - We now clearly state that the panoptic backbone, scene-graph idea, and T5-based decoder are **existing architectures**; our contribution lies in:
>   - Their **specific composition and constraints**, and
>   - The **ablation and robustness analysis** (e.g., structured bottleneck vs. direct ViT→T5, triplet filtering, ordering effects).
>
> This reframing avoids overstating algorithmic novelty and emphasizes the integration and analysis aspects.
>
> ---
>
> ## 5. Human evaluation details
>
> **Concern (paraphrased):** The human study is not described in enough detail to judge its validity.
>
> **Response / changes.**
>
> - The appendix subsection *“Human Evaluation Protocol”* now specifies:
>   - **Three annotators**,
>   - **150 validation images** (balanced by scene density),
>   - **Three 1–5 Likert scales**: Fluency, Relevance, and Spatial correctness,
>   - **Randomized and anonymized** model outputs for each image.
> - We report **Krippendorff’s α = 0.62** (averaged across dimensions) as an inter-rater reliability measure.
> - The main paper briefly references this protocol when summarizing human results, so readers can better assess the **statistical robustness** of the human evaluation.
>
> ---
>
> We sincerely appreciate your feedback; it helped us improve clarity, positioning, and evaluation in the revised manuscript.

---

### Official Review · Reviewer_iJTU · 2025-11-01

**Soundness:** 2
**Presentation:** 1
**Contribution:** 1
**Rating:** 2
**Confidence:** 4

**Summary:**

This paper designs a new framework that uses an intermediate graph structure to improve the accuracy of spatial information in VLM predictions.

**Strengths:**

The main idea makes sense. The experiments are small, but they still show that their approach, using a smaller T5 model, managed to beat larger models.

**Weaknesses:**

This paper has two very significant flaws.
1. First, the writing quality is poor and suggests a rushed submission. The figures are crude, typos are frequent (like line 307), and the text is confusing. They even forgot to describe the loss function, which is a critical omission.
2. Second, the experiments are far too small to be convincing. The conclusions cannot be trusted because the scale is inadequate: the relation set is tiny (12 categories), the testset is only 50 images, and the training set is just 5000 images from one source. This is not enough to prove the method's generalizability.

Besides, I think there's another major problem: the paper relies on similarity-based metrics (e.g., BLEU), which are ill-suited for this task. These metrics fail to capture the factual correctness of critical spatial information, which is the core contribution. The reported gains may simply reflect overfitting to linguistic patterns rather than true improvements in spatial reasoning. A more robust, fact-centric evaluation is required.

**Questions:**

Please refer to the issues detailed in the Weakness part.

---

> ### Author Response · Authors · 2025-12-04
> **Author response to Reviewer ijTU**
>
> Thank you for your detailed review. We address your main concerns and describe the corresponding revisions.
>
> ---
>
> ## 1. Writing quality, figures, and loss description
>
> **Concern:** The draft felt rushed: crude figures, typos, confusing exposition, and missing loss description.
>
> **Response / changes.**
>
> - We edited the **Introduction**, **Methodology**, and **Experimental Setup** for clarity and removed many redundant phrases, with more direct definitions of HPSG, triplets, and NRDS.
> - We performed a thorough **proofreading pass**, fixing typographical and formatting issues (including the specific line you pointed out).
> - All main figures were **redrawn** with clearer layouts and consistent styling.
> - We now **explicitly describe the training objective**: the T5 decoder is trained with token-level cross-entropy loss with label smoothing, optimized with AdamW. Optimization and decoding details are summarized in the appendix subsection *“Reproducibility and Implementation Details”*.
>
> ---
>
> ## 2. Experimental scale and generalizability
>
> **Concern:** The relation set, training set, and test set are too small to support strong conclusions about generalization.
>
> **Response / changes.**
>
> - We clarify that we use the **standard Cityscapes fine split** (2,975 train, 500 val, 1,525 test images). The previous draft did not state these numbers prominently.
> - The **12-predicate relation set** (left/right, in front/behind, on/inside, near/adjacent, overlaps/occludes, part_of/contacts) is deliberately compact and follows prior scene-graph work. In the **“Ablations and Analysis”** section and appendix, we now include a **sensitivity study** varying both the predicate inventory and the **triplet budget $K$**, showing that the improvements are robust to these choices.
> - The **50-image BDD100K subset** is now clearly presented as a **zero-shot transfer probe**, not as a full benchmark. Its purpose is to test whether the HPSG→triplet bottleneck gives any robustness under a moderate distribution shift.
> - In **“Limitations and future work”**, we explicitly note that our experiments are **restricted to urban driving** and that broader, multi-domain studies are required to fully establish generalizability.
>
> We hope this makes the scope and strength of our claims more precise.
>
> ---
>
> ## 3. Metrics and factual/spatial correctness
>
> **Concern:** BLEU-style similarity metrics do not capture factual spatial correctness; reported gains may just reflect linguistic overlap.
>
> **Response / changes.**
>
> We agree that text-only metrics are not sufficient, and we have strengthened the evaluation to focus more directly on grounding:
>
> - Besides BLEU/ROUGE/METEOR/CIDEr/SPICE/BERTScore, we now emphasize:
>   - **CHAIR-s/i** and **Entity-Precision** to quantify hallucinated vs. correctly grounded entity mentions.
>   - Our **Narrative Relevance Detection Score (NRDS)**, described in detail in the **“Evaluation Metrics”** section and a dedicated appendix subsection.
> - NRDS is explicitly **instance-level and fact-centric**:
>   - It combines **detection correctness** ($\mathrm{DetAcc}_j$), **class-dependent narrative importance** ($\mathrm{NarrImport}_j$), and **instance–paragraph alignment** ($\mathrm{ParaAcc}_j$ based on CLIP and noun-phrase matching), aggregated and normalized per image.
>   - Unlike CLIPScore, which uses a single image–paragraph similarity, NRDS scores each detected instance, so models must both detect and describe important objects accurately to score well.
> - We also added a **blind human evaluation** (appendix *“Human Evaluation Protocol”*). Annotators rate **Fluency**, **Relevance**, and **Spatial correctness** given only the image. PG-VLM is preferred over all baselines on all three criteria, including spatial correctness.
>
> Crucially, PG-VLM improves not only on BLEU/CIDEr but also on **CHAIR, Entity-Precision, NRDS, and human spatial ratings**, indicating that the gains reflect better factual grounding rather than mere stylistic alignment.
>
> ---
>
> ## Summary
>
> In response to your review, we have:
> 1. **Improved the writing and figures** and **made the loss and optimization setup explicit**, addressing the impression of a rushed submission.
> 2. **Clarified the experimental scale**, added **predicate and triplet-budget ablations**, and stated the **limitations** of the current domain and dataset coverage more clearly.
> 3. **Strengthened the evaluation** with **grounding-oriented metrics (CHAIR, NRDS)** and a **blind human study**, so that improvements track spatial and factual correctness, not just lexical similarity.
>
> We appreciate your critical feedback; it led to a clearer and more rigorously evaluated version of the paper.

---

### Official Review · Reviewer_xYbm · 2025-11-01

**Soundness:** 1
**Presentation:** 1
**Contribution:** 1
**Rating:** 0
**Confidence:** 4

**Summary:**

This paper approaches the task of detailed image captioning on the Cityscapes dataset. The proposed model first builds a graph from panoptic segmentation outputs and their spatial positioning relationships. From graph triplets, a T5 generates the final detailed caption for the urban scene. This paper presents results on this task using pseudo-labels and compares them to those of other state-of-the-art image captioning models.

**Strengths:**

This review evaluates the paper's quality based on the following criteria: task relevance, related work, technical novelty, technical correctness, experimental validation, writing and presentation, and reproducibility. Each aspect is discussed and highlighted as a strength or a weakness in the sections below.
-    **Relevance of the task:**  Detailed Image Captioning is a highly relevant problem for the machine learning community.

**Weaknesses:**

-    **Reproducibility and Implementation Details:** It is not indicated whether the source code will be released, and it's not included as part of the submission.
-    **Related Work and Technical Novelty:** The Related Work section does not adequately contextualize the contributions. It is not clear how the proposed method addresses the limitations of current detailed image captioning methods.
-    **Writing and Presentation:** This paper is not easy to read. It lacks sections on the metric formulation. Its organization does not allow the reader to adequately understand the problem/motivation, its prevalence in current state-of-the-art methods, the proposed methodology, how this methodology solves the research gap, and finally, how well the experiments support it.
-    **Experimental Validation and Technical Correctness:** The empirical validation of this paper is not clear from the dataset used. According to the paper description (Lines 240–244), it utilizes the panoptic annotations of Cityscapes for training and pseudo-labels for the detailed image captioning task, generated using the same T5 model. Are these pseudo-labels also used for evaluation and comparison with previous state-of-the-art methods? This experimental choice will result in an obviously biased validation towards the outputs of the pretrained model, which puts the proposed method at an evident advantage against any other model not based on this generator.

**Questions:**

1.	Will the source code and pretrained models be released to support reproducibility? If so, what is the reason for not including them in the supplementary material?
2.	How does the method improve upon or differ from existing detailed image captioning approaches?
3.	Can the paper clarify the motivation, the gap in prior work, and how the method addresses it?
4.	What metrics are used for evaluation, and can their formulation be clarified?
5.	Are the pseudo-labels used for both training and evaluation? If so, how is evaluation bias avoided?
6.	How does the use of pseudo-labels affect fairness in comparison with other models?

---

> ### Author Response · Authors · 2025-12-04
> **Author response to Reviewer 1**
>
> Thank you for your careful read and constructive feedback. Below we address your main comments and summarize the changes in the revised paper.
>
> ---
>
> ## 1. Clarification and justification of NRDS
>
> **Concern (paraphrased):** The definition of NRDS and its relation to CLIP-based metrics were not fully clear.
>
> **Response / changes:**
> - In **“Evaluation Metrics”** (subsection *Narrative Relevance Detection Score (NRDS)*) and the corresponding appendix subsection, we now give the **full mathematical definition** and define all components.
> - We add a short **comparison to CLIPScore**, emphasizing that NRDS (i) is computed at **instance level** and (ii) jointly uses **detection correctness**, **class importance**, and textual realization, so a model cannot score well by describing irrelevant or wrongly detected content.
> - We report **per-class NRDS** in the appendix, showing that rare but important classes benefit most from the HPSG→triplet bottleneck.
>
> ---
>
> ## 2. Pseudo-labeling protocol and fairness to baselines
>
> **Concern (paraphrased):** The pseudo-labeling scheme could bias results toward PG-VLM and disadvantage baselines.
>
> **Response / changes:**
> - The subsection **“Training Data, Pseudo-Labels, and Evaluation Protocol”** now describes the full pipeline: for each image we (i) build an HPSG and extract triplets, (ii) ask a **fixed local instruction model** (LLaMA-2-7B-Chat) to generate a paragraph only from these triplets, and (iii) remove any sentence mentioning entities not in the triplets.
> - These teacher paragraphs serve both as **training targets** for the T5 decoder and as **references** for *all* models (PG-VLM, BLIP-2, LLaVA-1.5, SpatialVLM) when computing automatic captioning metrics, so every model is evaluated against the same text.
> - To mitigate bias we highlight metrics that depend on **image–text grounding** rather than lexical overlap (CHAIR-s/i, Entity-Precision, NRDS) and we add a **blind human study** (appendix *“Human Evaluation Protocol”*) where annotators only see the image. PG-VLM is preferred over all baselines on fluency, relevance, and spatial correctness.
>
> ---
>
> ## 3. Reproducibility, ablations, robustness, and cost
>
> **Concern (paraphrased):** Implementation details, ablations, and runtime/robustness were not sufficiently specified.
>
> **Response / changes:**
> - The appendix subsection **“Reproducibility and Implementation Details”** now gives concrete settings for the software environment, the Mask2Former backbone, HPSG construction (predicate set, relation selection, **top-$k$ edges**), triplet serialization and **top-$K$ filtering**, and T5-Large training/decoding (including the post-checker used to re-rank beams).
> - In **“Ablations and Analysis”** and the appendix we add: (i) an ablation that removes the HPSG bottleneck and feeds ViT features directly to T5 (reducing captioning and grounding metrics and increasing hallucination); (ii) an ablation on **triplet filtering** (top-$K$ vs. all triplets), and (iii) an ablation on **triplet ordering** (layout→agents vs. random), all supporting the value of the HPSG→triplet bottleneck.
> - We further include **robustness and cost** results: NRDS under different occlusion levels and brightness/contrast perturbations, and a runtime/memory table with per-stage latency and VRAM on a single RTX 4090 (≈245 ms per image at 1024×512).
> - On OpenReview we now refer to tables and appendix material by **descriptive titles and subsection names** rather than LaTeX labels, so these references no longer appear as “??”.
>
> ---
>
> ## 4. Limitations, broader impact, and future work
>
> **Concern (paraphrased):** The limitations and broader impact of PG-VLM and NRDS were not discussed in enough detail.
>
> **Response / changes:**
> - We extended **“Limitations and future work”** and the appendix subsection **“Broader Impact and Risk Controls (Extended)”** to note that PG-VLM depends on both the panoptic backbone and the teacher LLM and that errors or biases in these components can propagate through the HPSG and triplets into the narrative.
> - We stress that our experiments are restricted to **urban driving** (Cityscapes and a BDD100K subset) and that testing the approach on other domains is important future work.
> - We explicitly mention that NRDS still relies on CLIP and heuristic phrase matching and outline follow-up work on more principled grounding metrics and alternative visual backbones, together with simple operational guidance (thresholding CHAIR/NRDS, human-in-the-loop checks, privacy filters, and evaluation under diverse conditions).
>
> ---
>
> In summary, the revision (i) clarifies and motivates NRDS, (ii) makes the pseudo-labeling and evaluation protocol explicit and symmetric across models, (iii) adds detailed implementation settings plus ablations, robustness, and runtime analyses, and (iv) strengthens the discussion of limitations and broader impact. Your feedback helped us improve clarity and reproducibility, and we thank you for it.

---

### Meta-Review · Area_Chair_ir7F · 2026-01-05

**Summary:**

The reviewers all agreed that this paper is not yet ready for publication. The primary reasons for this decision are poor writing quality, insufficient experimentation, and limited technical novelty.

**Reviewer Concerns:**

- Presentation and Writing: Despite the clarifications, the overall presentation quality remains below the conference standard. The manuscript suffers from structural issues that obscure the contribution and motivation.
- Experimental Analysis: While the authors explained the validity of their protocol, the comprehensiveness of the analysis is still insufficient.

**Reviewer Scores:**

I believe the reviewers will not change their scores, as the rebuttal primarily focused on clarifying experimental details rather than addressing fundamental weaknesses, such as writing quality, outdated benchmarks, and reproducibility.

---

### Decision · Program_Chairs · 2026-01-26

Reject